# Changes in Lipoprotein(a) Levels in People after ST Elevation Myocardial Infarction—The STEMI-Lipids Study

**DOI:** 10.3390/ijms242115531

**Published:** 2023-10-24

**Authors:** Caren Sourij, Faisal Aziz, Sarah Krappinger, Andreas Praschk, Thomas Metzner, Harald Kojzar, Andreas Zirlik, Tatjana Stojakovic, Dieter Pätzold, Dirk von Lewinski, Robert Zweiker, Hubert Scharnagl, Harald Sourij

**Affiliations:** 1Division of Cardiology, Medical University of Graz, Auenbruggerplatz 15, 8036 Graz, Austriaandreas.praschk@medunigraz.at (A.P.); dieter.paetzold@aon.at (D.P.);; 2Trials Unit for Interdisciplinary Metabolic Medicine, Division of Endocrinology and Diabetology, Medical University of Graz, Auenbruggerplatz 15, 8036 Graz, Austria; faisal.aziz@medunigraz.at (F.A.); sarah.krappinger@stud-medunigraz.at (S.K.);; 3Department of Medical Affairs, Eli Lilly GmbH, Erdberger Lände 26A, 1030 Vienna, Austria; 4Clinical Institute of Medical- and Chemical Laboratory Diagnostics, University Hospital Graz, 8036 Graz, Austria; stojakovic@gmx.at (T.S.); hubert.scharnagl@medunigraz.at (H.S.)

**Keywords:** Lp(a), lipoprotein, acute myocardial infarction

## Abstract

Lipoprotein(a) (Lp(a)) is considered an independent risk factor for cardiovascular diseases. The plasma concentration of Lp(a) is largely genetically determined but varies over a wide range within the population. This study investigated changes in Lp(a) levels after an acute myocardial infarction. Patients who underwent coronary angiography due to an ST elevation myocardial infarction were enrolled (*n* = 86), and Lp(a) levels were measured immediately after the intervention, one day, two days, and at a post-discharge follow-up visit at 3 to 6 months after the acute myocardial infarction. Median Lp(a) levels increased from a median of 7.9 mg/dL (3.8–37.1) at hospital admission to 8.4 mg/dL (3.9–35.4) on the following day, then to 9.3 mg/dL (3.7–39.1) on day two (*p* < 0.001), and to 11.2 mg/dL (4.4–59.6) at the post-discharge follow-up (*p* < 0.001). Lp(a) levels were the lowest during the acute myocardial infarction and started to increase significantly immediately thereafter, with the highest levels at the post-discharge follow-up. The moderate but significant increase in Lp(a) in people with acute myocardial infarction appears to be clinically relevant on an individual basis, especially when specific Lp(a) cut-off levels are supposed to determine the initiation of future treatment. Hence, a repeated measurement of Lp(a) after myocardial infarction should be performed.

## 1. Introduction

Worldwide, more than 7 million people experience acute coronary syndrome (ACS) each year, including ST segment elevation myocardial infarction (about 30%) and non-ST segment elevation myocardial infarction (70%) [1]. Approximately 5% of those people die within days after hospitalisation, while 18% experience cardiovascular death, myocardial infarction or stroke within 4 years after the event [1,2]; however, modifiable risk factors, like smoking, hypertension, dyslipidaemia, diabetes mellitus, and obesity, are already identified and being treated.

One marker of cardiovascular risk, lipoprotein(a) (Lp(a)), attracts increasing interest, mainly because of treatment options being imminently available [3,4]. Similar to LDL particles, the Lp(a) core consists of triglycerides and cholesteryl esters, which are surrounded by phospholipids, unesterified cholesterol, and a single copy of apolipoprotein B-100 (apoB). In Lp(a), apoB is covalently linked to apolipoprotein(a) [apo(a)] via a disulfide bond [5]. Apo(a) contains repeated Kringle structures that are comparable to those of plasminogen but with extensive heterogeneity, not only within populations but also within one individual, where two different-sized apo(a) isoforms can be found in more than 80% of individuals [6].

Lp(a) promotes cardiovascular disease probably through various mechanisms. Lp(a) enters the arterial wall and undergoes oxidation after entry, favouring inflammation, atherosclerotic processes, calcification, and thrombus formation [5,6].

The growth of knowledge regarding Lp(a) as an atherosclerotic cardiovascular disease (ASCVD) risk factor has resulted in calls for its universal routine measurements in individuals with a strong family history of premature ASCVD by various cardiologic societies including the European Society of Cardiology and the Canadian Cardiovascular Society [7,8]. In particular, in people with ASCVD who have already well-controlled, established cardiovascular risk factors, Lp(a) remains to be an important determinant of residual cardiovascular risk, since for every combination of cardiovascular risk factors, elevated Lp(a) levels further increase the risk for future events [9,10].

As previous data have shown that Lp(a) levels are mostly genetically determined, current guidelines suggest measuring it at least once in adults [5,8,11]. However, it remains to be elucidated whether an acute event such as myocardial infarction can alter Lp(a) levels, potentially requiring further assessments after hospitalisation for an acute event.

In this prospective clinical observational study, we investigated changes in Lp(a) levels in people with ST elevation myocardial infarction at four timepoints, starting in the catheter lab.

## 2. Results

In total, 86 individuals (25 female) with acute myocardial infarction (35.9% anterior myocardial infarction; 64.1% non-anterior myocardial infarction) were enrolled between June 2019 and April 2022 at the Division of Cardiology, Medical University of Graz, Austria. Briefly, 3 participants died during the follow-up and 21 participants were not willing or unable to return for their final follow-up visit, which took place in a median of 115 days (IQR: 94–177 days) after acute myocardial infarction.

Median Lp(a) at admission (visit 1) was 7.9 mg/dL (3.8–37.1), and median LDL cholesterol and HDL cholesterol were 130 mg/dL (102–154) and 47 mg/dL (41–57), respectively (Table 1).

Median Lp(a) levels increased significantly by 4.9% to a median of 8.4 mg/dL (3.9–35.4) on the first day following MI, 9.3 mg/dL (3.7–39.1) on the second day, and 11.2 mg/dL (4.4–59.6) at the post-discharge follow-up. Table 2 and Figure 1 display the percentage changes over the course of the study. In total, 62 participants (72%) had Lp(a) within the normal range (<30 mg/dL) on admission. The Lp(a) category of two participants changed from normal (<30 mg/dL) to above >30 mg/dL at the post-discharge follow-up, while no participants’ category changed from the abnormal to normal category during that time. Women showed significantly higher Lp(a) levels at visit 4 compared with men (*p* < 0.001). When missing data in follow-up visits were imputed (using the MICE (Multiple Imputation by Chained Equation) method), the results remained similar, showing a significant increase over time in Lp(a) levels.

The median total cholesterol decreased significantly from the time of admission to post-discharge follow-up (195 [169–228] to 118 [108–142], *p* < 0.001). Median HDL cholesterol decreased in the first 48 h after myocardial infarction to 40 mg/dL (35–49) (*p* < 0.001), but then increased to a median level of 50 mg/dL (41–60) at post-discharge follow-up. The median triglyceride levels increased significantly from 74 mg/dL (53–101) at visit 1 to 134 mg/dL (97–168) and 126 mg/dL (99–169) on the following 2 days, respectively, and then decreased to a level of 97 mg/dL (75–124) at the post-discharge follow-up.

The median LDL cholesterol decreased to 98 mg/dL (75–125), 88 mg/dL (65–112), and 49 mg/dL (40–61) on day one, day two, and post-discharge follow-up, respectively.

## 3. Discussion

In this study, we observed a significant increase in Lp(a) levels by 31% from the day of acute myocardial infarction to the follow-up after hospital discharge. However, the clinical relevance of this increase in Lp(a) levels over time is more important. At baseline, 16 people (18.6%) had Lp(a) levels above the normal range (<30 mg/dL), and during the follow-up, 2 additional individuals demonstrated Lp(a) levels of >30 mg/dL, representing an additional 12.5% in the group with increased Lp(a) over time. We believe that this is a clinically relevant proportion, that would have been considered to have normal Lp(a) levels, when assessed only at the time of the acute event. Moreover, the observed higher Lp(a) concentrations in our study in women, particularly at the post-discharge follow-up, are in line with previous findings showing higher Lp(a) in women [12,13]. Hence, these data suggest that clinicians should specifically pay attention to the management of this risk factor during the follow-up of women after myocardial infarction.

Our findings regarding an increase in Lp(a) are also in line with those of a previous study showing higher Lp(a) levels in people 6 months after acute myocardial infarction [14]. Previous data indicate that Lp(a) levels may decrease during an acute myocardial infarction event, which is potentially similar to the decreases in Lp(a) levels that have been observed in individuals experiencing sepsis and severe burns [15]. Hence, in our study, the observed increase in Lp(a) could merely be a normalisation of acutely reduced Lp(a) levels rather than an actual increase. However, as previous research has also suggested that statins might increase Lp(a) levels [10,16], the rise might also be explained by the initiation of high-intensity statin treatment immediately after acute myocardial infarction.

In our study, participants displayed a significant decrease in LDL cholesterol levels to a mean LDL cholesterol of 49 mg/dL (40–61) at post-discharge follow-up, representing good implementation of lipid-lowering guidelines for secondary prevention after myocardial infarction. However, the risk for another cardiovascular event still remains increased with increasing Lp(a) levels.

One limitation of our study is the lack of information on Lp(a) levels before acute myocardial infarction. Hence, we cannot conclude from our data whether an acute event decreases Lp(a) levels or whether the acute event leads to a sustainable rise in Lp(a). Another limitation of our study is that it was performed during the COVID-19 pandemic and during regular working hours only, potentially introducing some bias in patient selection. Moreover, we have complete data only for the first three visits, as 21 patients did not come back for the final visit, also mostly due to the pandemic situation. Since the rise in Lp(a) levels was already observed within the first days after acute myocardial infarction, reaching statistical significance already at visit 3, our data suggest that the finding is unlikely due to missing data. Moreover, we performed a sensitivity analysis by imputing the missing data using the MICE method, and the results remained the same. Whether patients with increasing Lp(a) levels face higher risks of future cardiovascular events cannot be answered in our trial given the limited number of patients.

Our results are of clinical relevance on an individual basis, especially when specific Lp(a) cut-off levels are supposed to determine the initiation of future novel treatment options including antisense oligonucleotide and small interfering RNA, both targeting apo(a) production in hepatocytes and leading to a pronounced Lp(a) reduction of 71–97% [4,17,18]. Ongoing outcomes trials will elucidate if this significant Lp(a) reduction also translates into cardiovascular event reduction. Of the currently available lipid-lowering drugs, PCSK9 inhibitors and inclisiran were demonstrated to reduce Lp(a) levels by up to 25% while bempedoic acid only displayed minimal effects on Lp(a) lowering [19,20,21].

However, more studies on repeated measurements of Lp(a) after myocardial infarction are needed to investigate changes in Lp(a) levels during and after cardiovascular events, as shifts in risk profiles are likely to occur, potentially influencing treatment initiation [5].

## 4. Materials and Methods

This is a prospective observational study that investigates lipoprotein levels over a period of more than 3 months after ST elevation myocardial infarction. The STEMI-lipids study was approved by the Ethics committee of the Medical University of Graz, Austria (EK 31-024ex18/19). The study was carried out in accordance with the 1964 Declaration of Helsinki and adhered to the guidelines of Good Clinical Practice (ICH GCP E6). Each participant signed an informed consent prior to enrolment.

The study enrolled 86 consecutive patients who were admitted to the University Hospital of Graz for acute ST elevation myocardial infarction and were willing to participate in the trial. The trial enrolment lasted almost 3 years due to the following reasons: (i.) as the first sample had to be collected in the catheter lab and sent for analyses, enrolment took place only during the daytime and not on weekends; (ii.) in particular, during the first months of the COVID-19 pandemic, enrolment was paused. Blood samples were collected at day 1 (upon admission, before intervention; PCI), day 2 (in the morning), and day 3 (in the morning), as well as during a follow-up visit scheduled more than 3 months after myocardial infarction, reflecting the final hospital related cardio-vascular (CV) risk assessment, adaption of medical therapy, and subsequent dismissal procedures to peripheral patient-centred care for these very-high-CV-risk patients. The first three visits were performed while the patients were still in the hospital.

Lp(a) was determined using an immunturbidimetric assay (TinaQuant) from Roche Diagnostics (Mannheim, Germany). Total cholesterol, triglycerides, and HDL cholesterol were measured using enzymatic assays from Roche Diagnostics. LDL cholesterol was calculated with the Friedewald formula. Troponin T was measured using a high-sensitivity electrochemiluminescence immunoassay from Roche Diagnostics (Basel, Switzerland) on a Roche Cobas c System analyser.

## 5. Statistical Analysis

The statistical analysis was performed in Stata (version 17.0) and RStudio (2023.06.0 + 421). Continuous variables were presented as mean ± standard deviation (SD) or median with interquartile range (IQR) if not normally distributed. Categorical variables were presented as frequencies with their percentages (%). The change in lipoproteins over visits was assessed using the linear mixed-effects model and coefficients with corresponding 95% confidence intervals (CIs) were reported as percentage changes from the time of admission to subsequent visits. As some data were missing for lipoproteins at follow-up visits, we imputed the data using the MICE method and reanalysed the data.

## Figures and Tables

**Figure 1 ijms-24-15531-f001:**
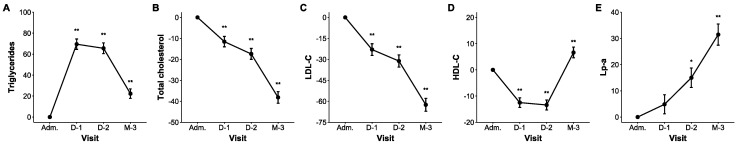
(**A**–**E**) Mean percentage change in lipoproteins over time. * *p* < 0.05, ** *p* < 0.001.

**Table 1 ijms-24-15531-t001:** Descriptive statistics of categorical and continuous variables.

Characteristic	
Age at admission (years), median (IQR)	61 (55–70)
BMI (kg/m^2^), median (IQR)	27.2 (25.0–30.7)
Triglycerides (mg/dL), median (IQR)	74 (53–101)
Total cholesterol (mg/dL), median (IQR)	195 (169–228)
LDL-C (mg/dL), median (IQR)	130 (102–154)
HDL-C (mg/dL), median (IQR)	47 (41–57)
Lp(a) (mg/dL), median (IQR)	7.85 (3.70–37.10)
Maximum CK (U/L), median (IQR)	1149 (575–2107)
Maximum Troponin T (pg/mL), median (IQR)	3773 (1259–6216)
Systolic blood pressure (mmHg), median (IQR)	126 (108–140)
Diastolic blood pressure (mmHg), median (IQR)	76 (66–83)
eGFR (ml/min/1.73 m^2^), median (IQR)	83.93 (69.48–96.53)
Sex	
Men, *n* (%)	61 (70.9)
Women, *n* (%)	25 (29.1)
BMI	
Underweight (<18.5), (%)	1.2
Normal range (18.5–24.9), (%)	23.3
Overweight (25–29.9), (%)	50
Obesity class I (30–34.9), (%)	17.4
Obesity class II (35–39.9), (%)	6.9
Obesity class III (>40), (%)	1.2
Diabetes (%)	18.8
Arterial hypertension (%)	75.3
Current smoker (%)	40.5
Past smoker (%)	21.4
eGFR categories (%)	
>90	36.1
<90	52.3
≤60	3.5
≤45	7.0
≤30	1.2
**Medication at hospital discharge**	
Any statin (%)	98.8
Simvastatin (%)	1.2
Atorvastatin (%)	96.4
Rosuvastatin (%)	2.4
Ezetimibe 10 mg (%)	8.5
PCSK9 inhibitor (%)	0.0
ASA (%)	96.4
Ticagrelor (%)	53.0
Prasugrel (%)	34.9
Clopidogrel (%)	10.8
ACE inhibitors/ARB (%)	89.2
Betablocker (%)	89.2
MRA (%)	29.0
NOAC (%)	7.2
OAC (%)	0.0

BMI: body mass index; HDL-C: high-density lipoprotein cholesterol; LDL-C: low-density lipoprotein cholesterol; Lp(a): lipoprotein(a); CK: creatinine kinase; eGFR: glomerular filtration rate; PCSK9: proprotein convertase subtilisin/kexin 9; ASA: acetylsalicylic acid; ACE: angiotensin-converting enzyme; ARB: angiotensin II receptor blocker; MRA: mineralocorticoid receptor antagonist; NOAK: non-vitamin K antagonist oral anticoagulants; OAK: oral anticoagulants.

**Table 2 ijms-24-15531-t002:** Generalised linear mixed-effects model of percentage change from admission.

Outcomes	Admission	Day 1		Day 2		Post-Discharge Follow-Up	
	Median (IQR)	Median (IQR)	∆ (95% CI)	Median (IQR)	∆ (95% CI)	Median (IQR)	∆ (95% CI)
Triglycerides (mg/dL)	74 (53–101)	134 (97–168)	**69.5 (54.2 to 86.3) ****	126 (99–169)	**65.5 (50.1 to 82.6) ****	97 (75–124)	**22.3 (10.2 to 35.8) ****
Total cholesterol (mg/dL)	195 (169–228)	172 (142–204)	**−11.6 (−15.7 to −7.2) ****	161 (134–194)	**−17.4 (−21.5 to −13.2) ****	118 (108–142)	**−38.1 (−41.4 to −34.8) ****
LDL-C (mg/dL)	130 (102–154)	98 (75–125)	**−23.0 (−29.0 to −16.5) ****	88 (65–112)	**−31.2 (−36.7 to −25.2) ****	49 (40–61)	**−62.5 (−65.7 to −59.1) ****
HDL-C (mg/dL)	47 (41–57)	40 (36–49)	**−12.5 (−15.6 to −9.3) ****	40 (35–48)	**−13.4 (−16.6 to −10.1) ****	50 (41–60)	**6.6 (2.5 to 11.0) ***
Lp(a) (mg/dL)	7.9 (3.8–37.1)	8.4 (3.9–35.4)	**4.9 (−2.2 to −12.4)**	9.3 (3.7–39.1)	**15.0 (7.0 to 23.5) ***	11.2 (4.4–59.6)	**31.4 (21.6 to 42.0) ****

∆: Mean percentage change from the values measured at admission. Changes provided in bold represent significant changes compared with admission. * *p* < 0.05, ** *p* < 0.001.

## Data Availability

Data are available upon reasonable request to the corresponding author of the study.

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
