# Peer review of "Changes in Lipoprotein(a) Levels in People after ST Elevation Myocardial Infarction—The STEMI-Lipids Study"

_ijms, 2023, doi:10.3390/ijms242115531_

Round 1

Reviewer 1 Report

This manuscript describes changes in Lp(a) levels after an ACS, which is a relatively new and interesting concept in clinical lipidology. Overall it is well written, yet the following issues should be addressed.

Major Comments

1.      It is not described the way these 86 patients were enrolled. This might hint towards a selection bias, which should be discussed as a clear limitation.

2.      The number of enrollees is quite small for the described period (2019-2022). This should be commented in the limitation section. Also, can you further explain why this happened? (budgetary issues, COVID-19 pandemic, drop-out rate..)

3.      ‘’ Lp(a) promotes cardiovascular disease probably through various mechanisms. Like

LDL particles, Lp(a) enters the arterial wall and tends to be oxidized after entry, creating highly immunogenic and proinflammatory oxidized phospholipids, with apo(a) potentiating atherosclerosis in the presence of lysine binding sites, allowing to accumulate in the vessel wall through antifibrinolytic effects mediated by inhibiting plasminogen activation 5,6’’  This section is poorly written. English and syntax needs to be rephrased

4.      Do you have clinical/outcome data concerning the patients who had their Lp(a) increased over follow up ? Did they suffer more events? If not add it to limitations.

Minor Comments

1.      Specify if troponin is high sensitive T or I (also describe the assay utilized)

2.      Table 1 Provide BMI categories in these way: overweight, class I, class II, class III obesity

3.      Table 1 abbreviations: use ASA instead of ASS, use ARB instead of AT2 receptor blocker, NOAC and OAC instead of NOAK and OAK

4.      Table 2 is poorly visualized, it needs editing to meet publication standards.

See comments in the attached file

Author Response

Major comments

  1. It is not described the way these 86 patients were enrolled. This might hint towards a selection bias, which should be discussed as a clear limitation.

86 consecutive patients, willing to participate were enrolled. We have added a sentence to the limitation section.

  1. The number of enrollees is quite small for the described period (2019-2022). This should be commented in the limitation section. Also, can you further explain why this happened? (budgetary issues, COVID-19 pandemic, drop-out rate..)

The trial enrollment lasted almost three years due to several reasons: i.) as the first sample had to be collected in the catheter lab and sent for analyses, enrollment took only place during day time and not on weekends, ii.) in particular during the first months of the COVID-19 pandemic, enrollment was paused.

Moreover, the principal investigator of the trial passed away during the conduct of the investigation and the study was taken over by the sub-PI.

  1. Lp(a) promotes cardiovascular disease probably through various mechanisms. Like LDL particles, Lp(a) enters the arterial wall and tends to be oxidized after entry, creating highly immunogenic and proinflammatory oxidized phospholipids, with apo(a) potentiating atherosclerosis in the presence of lysine binding sites, allowing to accumulate in the vessel wall through antifibrinolytic effects mediated by inhibiting plasminogen activation 5,6’’

This section is poorly written. English and syntax needs to be rephrased

Thank you for pointing this – we have rephrased this sentence.

  1. Do you have clinical/outcome data concerning the patients who had their Lp(a) increased over follow up? Did they suffer more events? If not add it to limitations.

We did not follow up these patients as the number of participants with increased Lp(a) was too small to create a meaningful analysis regarding outcome. We have added this to limitations.

Minor Comments

  1. Specify if troponin is high sensitive T or I (also describe the assay utilized)

TroponinT was measured using a high-sensitive electrochemoluminescence immunoassay from Roche Diagnostics (Basel, Switzerland) on a Roche Cobas c System analyser. We have added this information in the methods section.

  1. Table 1 Provide BMI categories in these way: overweight, class I,

 class II, class III obesity

We have modified the table as suggested.

  1. Table 1 abbreviations: use ASA instead of ASS, use ARB instead of AT2 receptor blocker, NOAC and OAC instead of NOAK and OAK

Done as suggested.

  1. Table 2 is poorly visualized, it needs editing to meet publication standards.

We have modified the table that should now comply with the journal formatting requirements.

Reviewer 2 Report

Caren Sourij et al. present a study with the aim to investigate changes in Lp(a) levels after an acute myocardial infarction. Although the topic is interesting and the manuscript well-written, some considerations need to be clarified.

1. Introduction: Describe the role of lipoprotein(a) in the cardiovascular risk assessment and how we can use it to improve the risk stratification. In addition, it is suggested to add the concept of “residual risk” in patients after myocardial infarction and the possible role of Lp(a).

Please, add the following references to improve the quality of the new sentences:
- Visseren FLJ, et al.; ESC Scientific Document Group. 2021 ESC Guidelines on cardiovascular disease prevention in clinical practice. Eur Heart J. 2021 Sep 7;42(34):3227-3337. doi: 10.1093/eurheartj/ehab484.
- Di Fusco SA et al. Lipoprotein(a): a risk factor for atherosclerosis and an emerging therapeutic target. Heart. 2022 Dec 13;109(1):18-25. doi: 10.1136/heartjnl-2021-320708.

2. Discussion: Authors should try to explain the reason for this significant increase in Lp(a) levers after acute myocardial infarction.

3. Discussion. In the final part of the discussion, it is suggested to describe the impact of current new therapies on Lp(a) levels as PCSK9 inhibitors, Inclisiran and Bempedoic acid.

Please, add the following references to improve the quality of the new sentences:
- Stefania Angela Di Fusco et al. Inclisiran: A New Pharmacological Approach for Hypercholesterolemia. Rev. Cardiovasc. Med. 2022, 23(11), 375. https://doi.org/10.31083/j.rcm2311375
- Yu Z, et al. Effect of Different Types and Dosages of Proprotein Convertase Subtilisin/Kexin Type 9 Inhibitors on Lipoprotein(a) Levels: A Network Meta-analysis. J Cardiovasc Pharmacol. 2023 Jun 1;81(6):445-453. doi: 10.1097/FJC.0000000000001419.
- Kim KA, Park HJ. New Therapeutic Approaches to the Treatment of Dyslipidemia 2: LDL-C and Lp(a). J Lipid Atheroscler. 2023 Jan;12(1):37-46. doi: 10.12997/jla.2023.12.1.37.

Author Response

  1. Introduction:

 Describe the role of lipoprotein(a) in the cardiovascular risk assessment and how we can use it to improve the risk stratification. In addition, it is suggested to add the concept of “residual risk” in patients after myocardial infarction and the possible role of Lp(a).

 Please, add the following references to improve the quality of the new sentences:

 - Visseren FLJ, et al.; ESC Scientific Document Group. 2021 ESC Guidelines on cardiovascular disease prevention in clinical practice. Eur Heart J. 2021 Sep 7;42(34):3227-3337. doi: 10.1093/eurheartj/ehab484.

 - Di Fusco SA et al. Lipoprotein(a): a risk factor for atherosclerosis and an emerging therapeutic target. Heart. 2022 Dec 13;109(1):18-25. doi: 10.1136/heartjnl-2021-320708.

We have now introduced the concept of residual risk and added the suggested references to the introduction section.

  1. Discussion: Authors should try to explain the reason for this significant increase in Lp(a) levers after acute myocardial infarction.

We discuss potential reasons for the observed increase of Lp(a) after an acute myocardial infarction, including a potential effect of the acute inflammation or the introduction of intensive oral lipid lowering treatment immediately after an acute myocardial infarction.

  1. Discussion. In the final part of the discussion, it is suggested to describe the impact of current new therapies on Lp(a) levels as PCSK9 inhibitors, Inclisiran and Bempedoic acid.

 Please, add the following references to improve the quality of the new sentences:

 - Stefania Angela Di Fusco et al. Inclisiran: A New Pharmacological

 Approach for Hypercholesterolemia. Rev. Cardiovasc. Med. 2022, 23(11),

  1. https://doi.org/10.31083/j.rcm2311375

 <https://doi.org/10.31083/j.rcm2311375>

- Yu Z, et al. Effect of Different Types and Dosages of Proprotein Convertase Subtilisin/Kexin Type 9 Inhibitors on Lipoprotein(a) Levels: A Network Meta-analysis. J Cardiovasc Pharmacol. 2023 Jun 1;81(6):445-453. doi: 10.1097/FJC.0000000000001419.

- Kim KA, Park HJ. New Therapeutic Approaches to the Treatment of Dyslipidemia 2: LDL-C and Lp(a). J Lipid Atheroscler. 2023 Jan;12(1):37-46. doi: 10.12997/jla.2023.12.1.37.

We have added a paragraph on the impact of novel lipid lowering drugs on Lp(a) levels and have cited the mentioned manuscripts.

Round 2

Reviewer 2 Report

The authors responded satisfactorily to my doubts and comments, congratulations